# Evaluating land-cover change and land subsidence in coal fire zones: Insights from multi-source monitoring

Desheng Xie[1]*, Fantao Zeng[1], Baozhu Liu[1], Quan Fang[1], Yongwei Dong[2], He Wu[2], Peng Liu[2], Ke Wang[2], Gensheng Li[3]

**1** The 9th Geological Brigade, Xinjiang Geological and Mineral Exploration and Development Bureau, Urumqi, China, **2** Xinjiang Urban Construction (Group) Co., Ltd., Urumqi, China, **3** Collaborative Innovation Center of Green Mining and Ecological Restoration for Xinjiang Mineral Re-sources, Xinjiang University, Urumqi, China

* 19190568622@163.com

**Data availability statement:** All relevant data are within the manuscript and its Supporting information files.

## Abstract

Coal fires are a significant environmental and geological threat, causing extensive ecological damage and loss of resources. Existing monitoring methods, though effective, have limitations in terms of precision and adaptability. This study integrates multiple monitoring techniques, including remote sensing, thermal infrared imaging, UAV based surveys, and field investigations, to assess the environmental and geological consequences of coal fires. The results indicate that: 1) Vegetation cover in the affected regions decreased from 0.43 to 0.38 between 2017 and 2024, with Fire Zone 1 showing minimal recovery and Fire Zone 3 exhibiting moderate improvement, though the overall low vegetation area expanded due to ongoing fires; 2) The Remote Sensing Ecological Index (RSEI) declined from 0.41 to 0.38, with poor-rated areas increasing significantly, particularly in Fire Zone 3, reflecting the ongoing degradation of ecological conditions caused by both fire activity and climate factors; 3) The Flat Spectral Shape Index (FSSI) showed stability, but fluctuations in the areas with moderate and high probability of exposed coal, especially in Fire Zone 3, pointed to the expanding fire footprint and worsening ecological degradation; 4) Surface subsidence varied considerably, with Fire Zone 3 experiencing the most severe subsidence, indicating unstable geological conditions exacerbated by fire activity. The study underscores the importance of continuous monitoring and effective disaster risk management strategies. Despite localized improvements in soil potassium levels, coal fires have caused significant declines in soil nutrients and moisture content. This research contributes to the development of more effective strategies for managing coal fire impacts and supporting ecological restoration efforts in affected regions.

## Introduction

Coal fires are a widespread and persistent environmental and geological hazard, affecting vast areas globally, particularly in coal-rich regions [1]. These fires, which can burn for decades,

**Funding:** This work was funded by the Basic Research Funds for Higher Education Institutions of Xinjiang Education Department (XJEDU2024J036), and Xinjiang Tianchi Talents (Young Doctor). The funders had role in study design, data collection and analysis, decision to publish, or preparation of the manuscript.

**Competing interests:** The authors have declared that no competing interests exist.

result in severe environmental degradation, including the destruction of ecosystems, loss of vegetation, and contamination of both soil and water resources [2]. Moreover, coal fires contribute significantly to the emission of greenhouse gases and pose serious health risks to nearby populations. The severity of these fires varies by region, with some of the most extensive and enduring coal fires found in countries such as China, India, and the United States. According to the 2019-2020 survey conducted by the Xinjiang Coal Fire. These sites covered a total burning area of 4.77 million m$^2$ [3]. Despite their far reaching and long-term environmental impacts, coal fires continue to present a formidable challenge for resource management and ecological restoration efforts [4].

In recent years, substantial progress has been made in the monitoring and management of coal fires [5]. Traditional methods such as ground based surveys and geological assessments have been complemented by advanced technologies, including remote sensing, thermal infrared imaging, and UAV (Unmanned Aerial Vehicle) surveys [6,7]. These technologies allow for more accurate and comprehensive monitoring of coal fire areas, enabling researchers and policymakers to assess fire extent, intensity, and environmental impact [8,9]. Various studies have focused on the development of models to predict coal fire behavior and its effects on local ecosystems, while others have explored mitigation techniques, such as fire suppression and ecological restoration [10]. Despite these advancements, the application of these technologies still faces challenges in terms of spatial coverage, temporal resolution, and the ability to· assess long-term ecological recovery in coal fire-affected areas.

However, current research on coal fire monitoring and management continues to be limited by several factors [11,12]. Existing methods often lack the precision required to monitor underground fire activity, and many studies focus on isolated methods, limiting the integration of data from different sources [12,13]. Additionally, while some monitoring techniques have shown promise in detecting fire zones, they often fail to provide real time, high resolution data that can be used for effective mitigation strategies [13,14]. It is susceptible to solar radiation, climatic conditions, surface type, and topographic variations [15,16]. These factors can lead to potential misjudgments in extracting thermal anomaly areas [17,18]. When the underground coal fire center exceeds 50 m in depth, the heat produced may not create significant surface thermal anomalies. This can result in missed detections and reduced accuracy in coal fire detection [19–21]. Thus, relying solely on single remote sensing information for coal fire detection often fails to meet practical needs [22,23]. This study addresses these gaps by integrating multiple monitoring techniques remote sensing, ground-based thermal infrared imaging, UAV surveys, and field investigations to provide a more comprehensive understanding of the environmental and geological impacts of coal fires [24,25]. The research also emphasizes the importance of multi method approaches in addressing the complexity of coal fire behavior and its long-term effects [26,27].

The primary objectives of this study are as follows: 1) To evaluate changes in vegetation cover and ecological recovery in coal fire-affected areas, focusing on FVC and RSEI dynamics. 2) To analyze the impact of coal fires on surface temperature and moisture conditions, particularly in Fire Zones 1 and 3. 3) To investigate the extent of exposed coal and the severity of coal fires using the Flat Spectral Shape Index (FSSI). 4) To assess the influence of coal fires on surface subsidence and land deformation in Fire Zones 1 and 3. Through this comprehensive approach, this research aims to contribute to the development of more effective monitoring and management strategies for coal fire-impacted regions [28,29].

## Materials and methods

### Materials

The study area is located in the southern mountainous region of Urumqi, China, within the northern temperate continental semiarid climate zone, characterized by an inter mountain basin climate (Fig 1). Summers are marked by unpredictable weather, including frequent rain

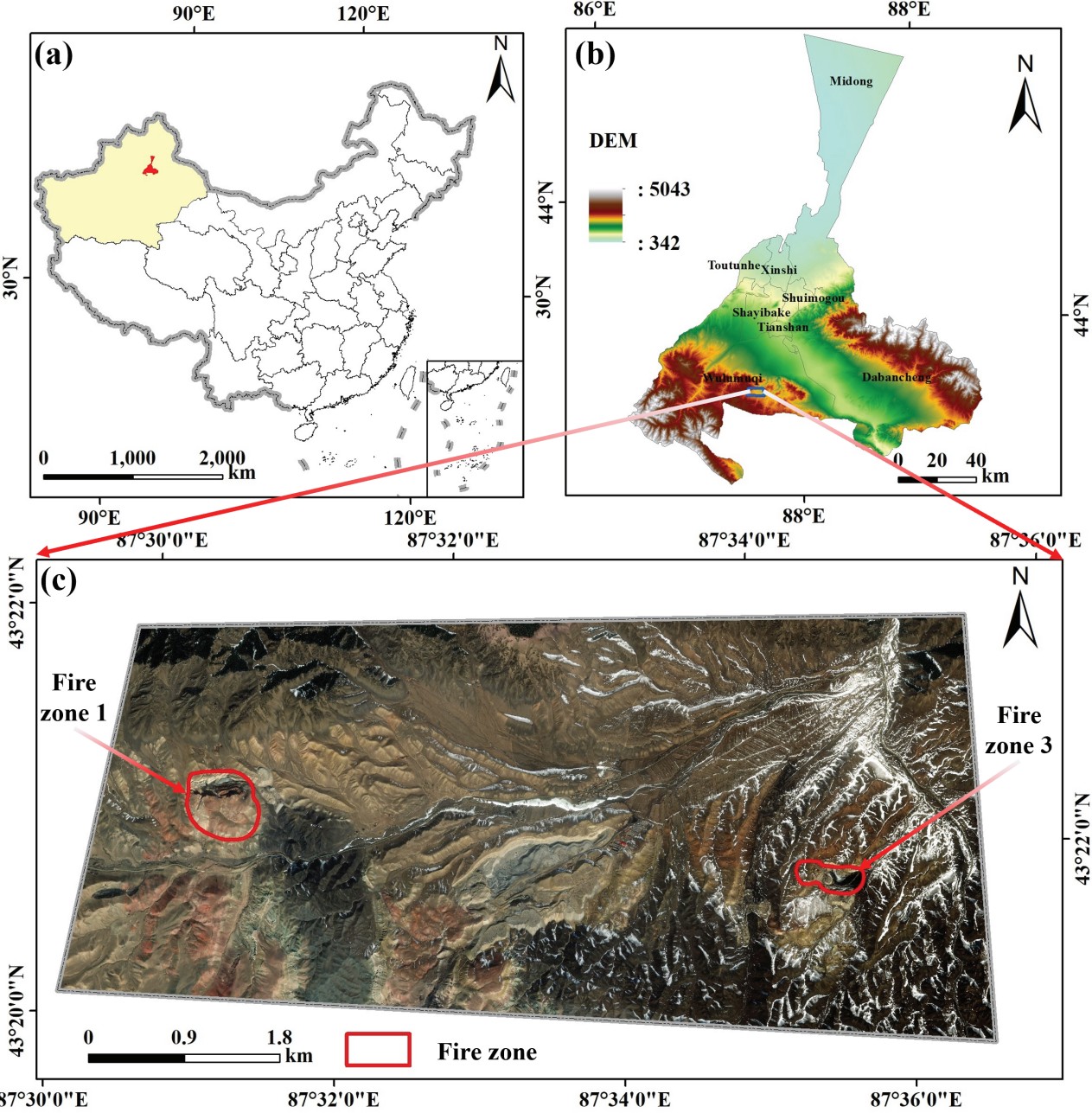

**Fig 1. Schematic diagram of the research area.** (a) is the location map of the research area. (b) is the topographic map of the research area. (c) is satellite imagery (Republished from Desheng Xie under a CC BY license, with permission from Desheng Xie, original copyright 2024. The source data was from NASA Earth Observatory).

and hail, while temperatures remain relatively cool, with a maximum of 25 °C and a minimum extreme low of –28 °C. The average annual temperature is 5 °C, with yearly precipitation ranging from 170 mm to 201 mm , and an average annual evaporation rate of 1882 mm . Wind speeds can reach up to 2.9 m/s.

Fire Zones 1 and 3 are located in the Mountain Basin at the northern foot of Tianshan Mountain (Fig 2). Fire Zone 1 is situated at an elevation of approximately 2200 m , with a maximum altitude difference of 100 m and a terrain slope ranging from 15 ° to 35 °. The pit in this zone extends about 300 m from east to west, 100  –  200 m from north to south, and is 50–100 m deep, with a significant amount of water at the bottom, ranging in elevation from 1948 m to 1948 m . In contrast, Fire Zone 3, with an altitude of around 1900 m , features a maximum elevation difference of 50 m and a slope of 10 ° to 15 °. The elliptical pit here is approximately 140 m in diameter and 20–30 m deep, with the bottom elevation ranging from 1847 m to 1848 m and containing a considerable amount of water.

## Methods

**Remote sensing ecological index.**  The remote sensing ecological index (RSEI) is a combination of the Land Surface Model (LSM), Normalized Differential Building and Bare Soil Index (NDBSI), and Land Surface Temperature (LST).

Landsat TM/ETM+/OLI/TIRS images from July 2017 to July 2024 were acquired from the United States Geological Survey (USGS) EarthExplorer platform. This paper uses satellite images of Landsat8 (https://landsat.gsfc.nasa.gov/satellites/landsat-8, Authorized). After obtaining these source data, we process and draw data on NASA Earth Observatory and PPT

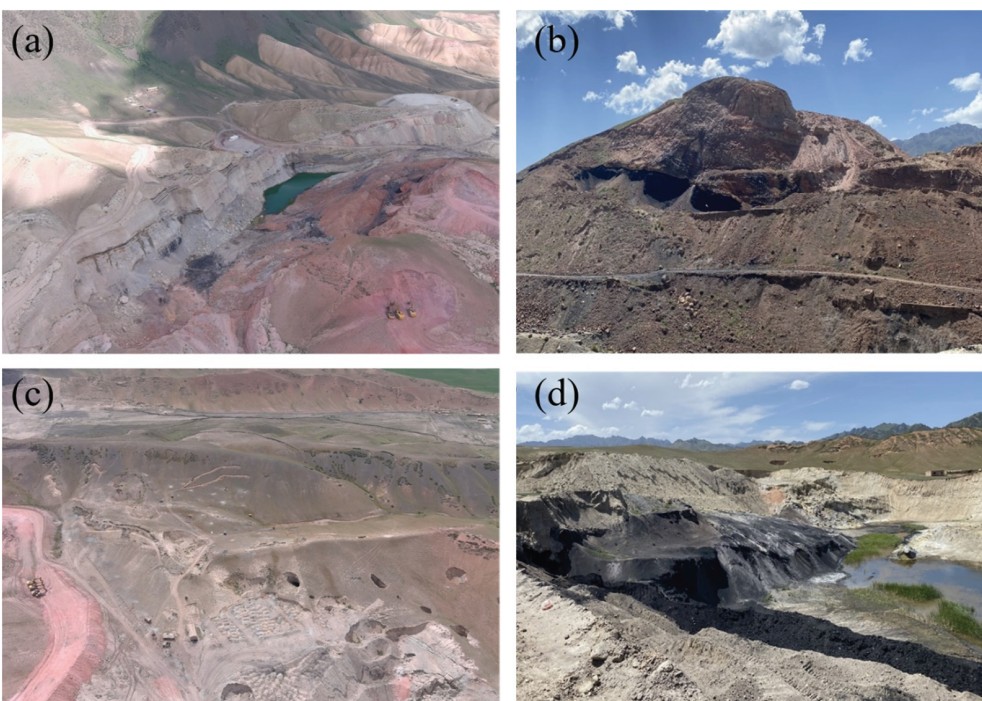

**Fig 2. Aerial view of Fire Zone 1 (a) and Fire Zone 3 (c); Mining disturbance geomorphology in Fire Zone 1 (b) and Fire Zone 3 (d).**

on our own computer. NASA Earth Observatory is a powerful cloud-based geospatial analysis platform that integrates massive remote sensing data (such as Landsat, Sentinel, etc.), high-performance computing, and a wealth of analytical tools to support environmental monitoring and research on a global scale. Its core advantage is that it can efficiently process petabyte data and complete complex calculations (such as global vegetation change analysis) without local hardware, which greatly improves research efficiency. At the same time, it provides free data access and open source community support, which is widely used in the fields of ecology, agriculture, disaster and climate, making remote sensing analysis more convenient and democratic (https://earthobservatory.nasa.gov/). All images were selected with cloud cover ≤10% to minimize atmospheric interference. The data pre-processing steps are carried out by ENVI 5.3, including radiometric calibration, atmospheric correction, geometric correction, Mosaic, and clipping. The FLAASH (Fast Line-of-sight Atmospheric Analysis of Spectral Hypercubes) module in ENVI was employed to correct for atmospheric scattering and absorption. Key parameters included Mid-latitude winter atmospheric model (consistent with the study area's semi-arid climate). Rural aerosol model with visibility set to 40 km. Water retrieval threshold was 1,130 nm. Images were co-registered to the WGS84/UTM Zone 45N coordinate system using ground control points (GCPs) from high-resolution Google Earth imagery (RMSE ≤ 0.5 pixels). Mosaics were generated using the "Seamless Mosaic" tool in ENVI, with histogram matching enabled to reduce inter-scene radiometric differences. Study area boundaries were clipped using a vector mask derived from the Xinjiang Geological Bureau's regional maps.

LSM is directly related to satellite–generated brightness, greenness, and humidity and can be used for ecological mapping and monitoring. LSM can be used for the fringe cap component of the Ecological Vulnerability Index. The humidity parameter is ex-pressed as land surface humidity, generated by the reflectance of Landsat TM and OLI images, as follows Eqs (1) and (2).

$$
\begin{aligned}
LSM_{OLI} = & \ k_{O1} * B_{Blue} + k_{O2} * B_{Green} + k_{O3} * B_{Red} \\
& + k_{O4} * B_{NIR} - k_{O5} * B_{NSWIII} - k_{O6} * B_{NSWIIX}
\end{aligned}
\tag{1}
$$

$$
\begin{aligned}
LSM_{TM} = & \ k_{T1} * B_{Blue} + k_{T2} * B_{Green} + k_{T3} * B_{Red} \\
& + k_{T4} * B_{NIR} - k_{T5} * B_{NSWIR1} - k_{T6} * B_{NSWIR\,2}
\end{aligned}
\tag{2}
$$

where $B_{Blue}, B_{Green}, B_{Red}, B_{sIR}, B_{swIR1}$, and $B_{swIR2}$ are the surface reflectance values of blue, green, red, near-infrared, and short-wave infrared bands of Landsat TM/ETM+ and OLI images respectively.

Normalized Difference Vegetation Index (NDVI) is the best measure of global greenness. Eq (3) can be easily measured by combining red and infrared bands.

$$
NDVI = \frac{B_{NIR} - B_{Red}}{B_{NIR} + B_{Red}}
\tag{3}
$$

where $B_{NIR}$ and $B_{Red}$ refer to the reflectance of the near-infrared and red bands.

NDBSI is generated from different band combinations of satellite images by soil index (SI) and building index (IBI), as follows Eqs (4), (5), and (6).

$$
SI = \frac{(B_{SWIRI} + B_{Red}) - (B_{Blue} + B_{NIR})}{(B_{SWIRI} + B_{Red}) + (B_{Blue} + B_{NIR})}
\tag{4}
$$

$$
IBI = \frac{\frac{2B_{SWIRI}}{B_{SWIRI} + B_{NIR}} - \left(\frac{B_{NIR}}{B_{NIR} + B_{Red}} + \frac{B_{Green}}{B_{Green} + B_{SWIRI}}\right)}{\frac{2B_{SWIRI}}{B_{SWIRI} + B_{NIR}} + \frac{B_{NIR}}{B_{NIR} + B_{Red}}}
\tag{5}
$$

$$NDBSI = \frac{IBI - SI}{2} \tag{6}$$

where $B_{Biue}$, $B_{Green}$, $B_{Red}$, $B_{NIR}$, and $B_{sWIR1}$ are the reflectance values of Landsat blue, green, red, near-infrared, and short-wave infrared bands.

$LST$ refers to heat, which can be easily determined from any satellite data of the tropics. Since Landsat data was used in this study, the thermal bands of Landsat 5 and 7 TM/ETM + sensors were used respectively, and the thermal bands of Landsat 8 OLI sensors were used. Finally, it is generated by Eqs (7), (8), and (9):

$$L_\lambda = \text{gain} * DN + \text{Bias} \tag{7}$$

$$T_b = \frac{K_2}{\ln\left(\frac{K_1}{L_\lambda + 1}\right)} \tag{8}$$

$$LST = \frac{T_b}{1 + \left(\frac{\lambda T_b}{P} \ln \varepsilon\right)} \tag{9}$$

where $L_\lambda$ is tropical. The gain is the thermal infrared gain value. All values can be found in the metadata file of the satellite data. And are tropical wavelengths (microns) and emissivity, respectively. The emissivity can be generated by the following Eq (10):

$$\varepsilon = mP + n \tag{10}$$

where $m$ is the soil radiance coefficient (0.004) and $n$ is the vegetation radiance coefficient (0.986). $P$ is the proportion of vegetation, which can be obtained by the following Eq (11):

$$P = \left(\frac{NDVI - NDVI_{\min}}{NDVI_{\max} - NDVI_{\min}}\right)^2 \tag{11}$$

To derive RSEI from the above four parameters, you first need to standardize all the factors, removing different dimension and range values. The four factors are normalized to homogenization and dimensionless on a scale of 0 to 1 , as follows Eq (12):

$$X_i = \frac{x_i - \min(x_i)}{\max(x_i) - \min(x_i)} \tag{12}$$

where $X_i$ is the standardized value of factor $i.x_i$ is the initial value of factor $i$.

**Fractional vegetation cover.** The Fractional Vegetation Cover (FVC) was calculated using the band computing tool, as shown in the following Eq (13):

$$\gamma = \frac{D_{ir} - D_r}{\delta_{ir} - \delta_r} - \frac{D_r - D_g}{\delta_r - \delta_g} \tag{13}$$

$$FVC = \frac{\gamma}{\gamma_{\max}} \tag{14}$$

where $D_{ir}, D_r$, and $D_g$ refer to the reflectance of near-infrared, red, and green bands in Eq (13) respectively; $\delta_{ir}, \delta_r, \delta_g$, are the center wavelengths of near-infrared, red, and green bands respectively. Eq (14) refers to the step difference index. The analysis of vegetation coverage can not only explore the temporal and spatial change characteristics of vegetation cover but also characterize the environment quality.

**Flat spectral shape index.** Since the reflectivity of water gradually decreases from the green band (B3) to the near-infrared band (B8), the band combination B8-B3 can be used to maximize the distinction between coal and water. In B8 and B3, the values of water are negative, while CMS, bare land, vegetation, and BRB are positive. Similarly, five types of land value can be absolute value index is built with abs $((\rho_{\text{NIR}} - \rho_{\text{Green}})/(\rho_{\text{NIR}} + \rho_{\text{Green}}))$ for normalization and symbolization. The index shown in Eq (15) was constructed and named Flat Spectral Shape Index (FSSI) [30], which highlights land cover types with flat spectral shapes, ranging from B2 to B3 and B3 to B8. The higher the FSSI, the flatter the spectral shape, as follows Eq (15):

$$\text{FSSI} = \frac{\exp\left(-\text{abs}\left(\frac{\rho_{\text{Blue}} - \rho_{\text{Green}}}{\rho_{\text{Blue}} + \rho_{\text{Green}}}\right)\right)}{1 + \text{abs}\left(\frac{\rho_{\text{NIR}} - \rho_{\text{Green}}}{\rho_{\text{NIR}} + \rho_{\text{Green}}}\right)} \tag{15}$$

where $\rho_{\text{Blue}}$ represents the blue band reflectance, $\rho_{\text{Green}}$ represents the green band reflectance, $\rho_{\text{NIR}}$ reflectance of the near-infrared wave band.

# Results

## Changes of forest vegetation cover

The results of the area proportion of each level after statistics are shown in Fig 3. The FVC changed from 0.43 in 2017 to 0.38 in 2024, showing a change of rising-stable-falling. The average FVC increased from 0.43 in 2017 to 0.43 in 2018. During this period, the relatively high vegetation cover area increased from 4.15 to 4.58 km$^2$. The FVC averages for Fire Zone 1 and 3 also increased from 0.03 and 0.16 in 2017 to 0.06 and 0.23 in 2018. Conversely, minimal changes were observed in other areas. The average FVC increased from 0.43 in 2018 to 0.41 in 2020. Unexpectedly, the average FVC of Fire Zone 1 and 3 increased from 0.06 and 0.23 to 0.11 and 0.25 .

The FVC decreased from 0.44 in 2021 to 0.38 in 2024. During this period, the low vegetation cover area increased from 6.09 to 10.27 km$^2$, and the medium vegetation cover area increased from 7.32 to 4.95 km$^2$. The relatively high vegetation cover area decreased from 4.54 to 3.43 km$^2$. The FVC for Fire Zone 1 and 3 decreased from 0.12 and 0.37 to 0.07 and 0.25.

## Changes of remote sensing ecological index

This classification illuminated the temporal evolution of the overall ecological security pattern. The statistical outcomes of the area proportions of each grade are presented in Fig 4. The trend of RSEI from 2017 to 2023 was generally consistent with that of FVC, However, there were also different changes in some specific years. The average RSEI declined from 0.41 in 2017 to 0.37 in 2021. During this time frame, the poor–rated area increased to 5.16 km$^2$ from 3.41 km$^2$, while the medium–rated area contracted to 7.60 km$^2$ from 8.70 km$^2$. Similarly, the good–rated area decreased to 2.72 km$^2$ from 3.60 km$^2$. The average RSEI decreased from 0.44 in 2021 to 0.38 in 2024. Finally, the average RSEI further decreased from 0.44 in 2021 to 0.38 in 2024. In this period, the poor–rated area increased to 7.34 km$^2$ from 5.16 km$^2$, and the medium–rated area decreased to 6.09 km$^2$ from 7.60 km$^2$. Conversely, the good–rated area diminished to 1.65 km$^2$ from 2.72 km$^2$.

LST increased significantly from 0.56 in 2017 to 0.74 in 2020, and then gradually recovered to 0.63 in 2024. LST in Fire Zone 1 decreased from 0.52 in 2017 to 0.58 in 2020 but suddenly increased to 0.24 in 2024. LST in Fire Zone 3 remained relatively high and fluctuated less throughout the observation period. The wetness of Fire Zone 1 was relatively stable during the

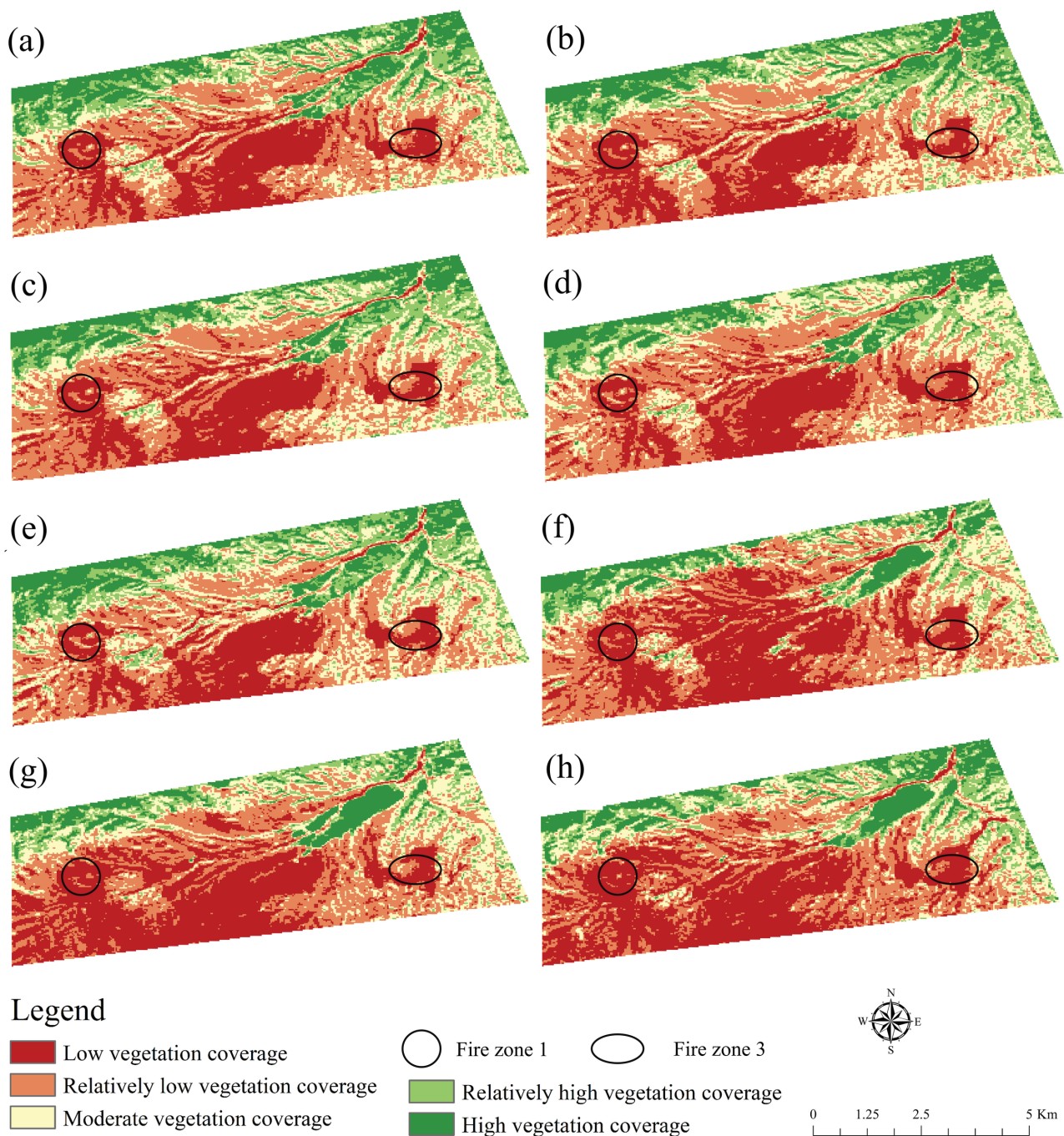

**Fig 3. The distribution of fractional vegetation cover.** (a), (b), (c), (d), (e), (f), (g), and (h) are 2017, 2018, 2019, 2020, 2021, 2022, 2023, and 2024, respectively (Republished from Desheng Xie under a CC BY license, with permission from Desheng Xie, original copyright 2024. The source data was from NASA Earth Observatory).

observation period. However, it showed a slow downward trend, especially reaching its lowest point at 0.41 in 2018 before recovering slightly. The wetness of Fire Zone 3 was relatively low and fluctuated greatly throughout the observation period.

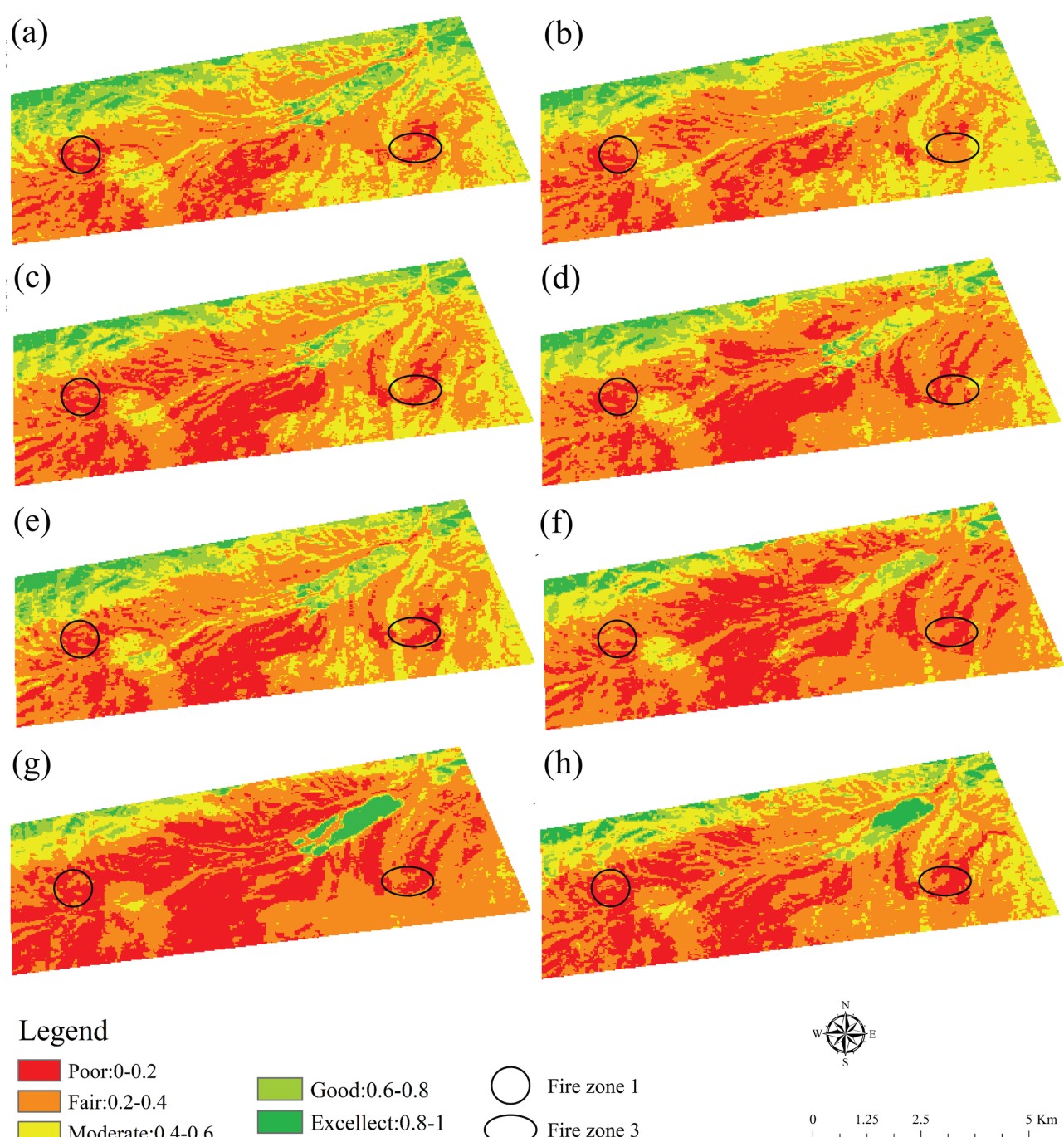

**Fig 4. The distribution of remote sensing ecological index.** (a), (b), (c), (d), (e), (f), (g), and (h) are 2017, 2018, 2019, 2020, 2021, 2022, 2023, and 2024, respectively (Republished from Desheng Xie under a CC BY license, with permission from Desheng Xie, original copyright 2024. The source data was from NASA Earth Observatory).

### Changes of flat spectral shape index

The statistical outcomes of the area proportions of each grade are presented in Fig 5. The FSSI fluctuated between 0.62 and 0.63 from 2017 to 2024, showing no clear long–term trend. During this period, the area of moderate probability of ex-posed coal also increased from 9.76

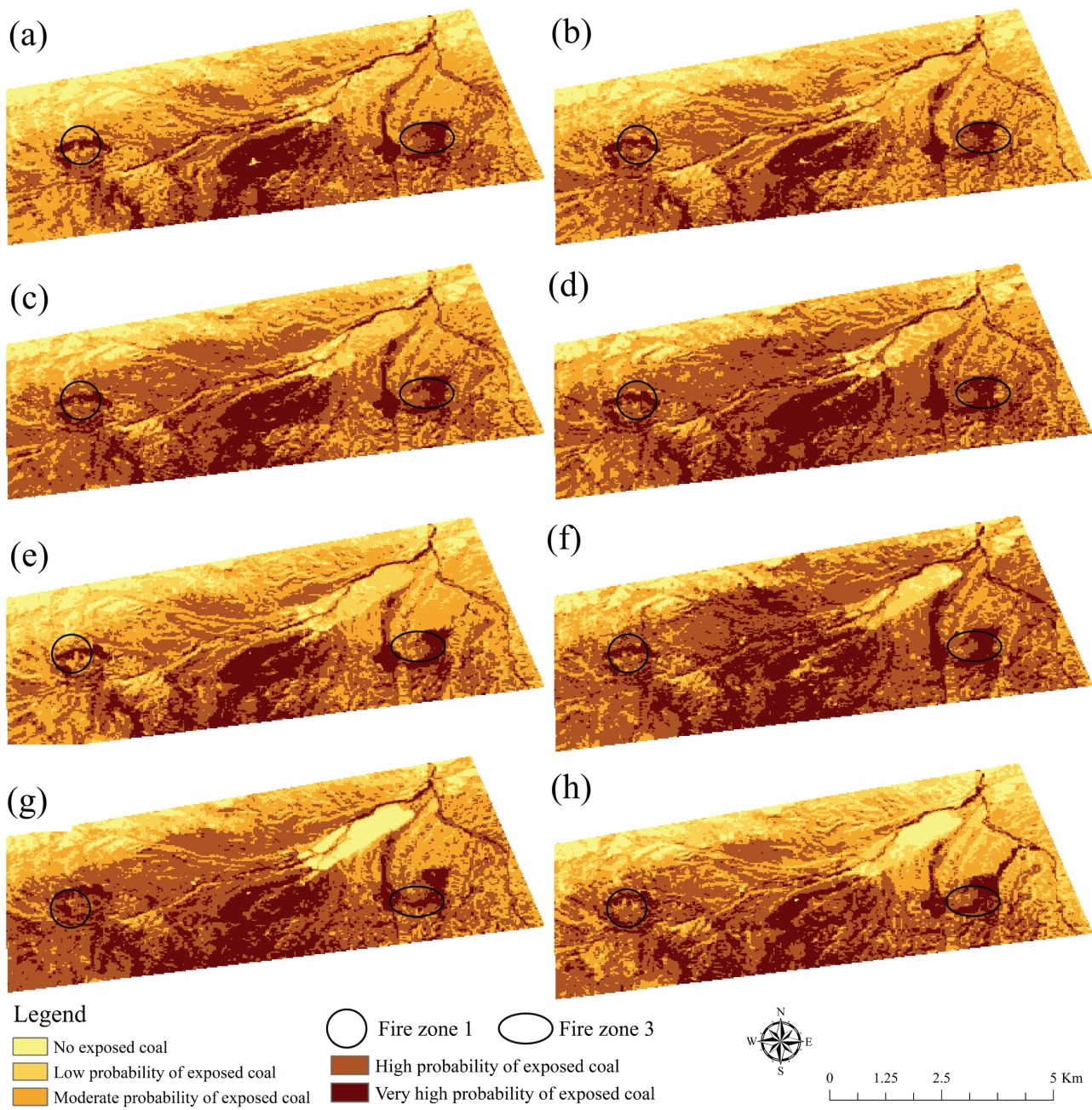

**Fig 5. The distribution map of flat spectral shape index.** (a), (b), (c), (d), (e), (f), (g), and (h) are 2017, 2018, 2019, 2020, 2021, 2022, 2023, and 2024, respectively (Republished from Desheng Xie under a CC BY license, with permission from Desheng Xie, original copyright 2024. The source data was from NASA Earth Observatory).

km$^2$ in 2017 to 6.67 km$^2$ in 2020 and then showed a fluctuating upward trend. However, the area with a high probability of exposed coal and a very high probability of exposed coal showed a fluctuating upward trend. FSSI of Fire Zone 1 still fluctuated at a low level. The FSSI of Fire Zone 3 fluctuates between 0.65 and 0.70.

## Changes of land subsidence

The statistical outcomes of the area proportions of each grade are presented in Fig 6. The overall ground subsidence, as well as the ground subsidence in Fire Zone 1 and 3, exhibited significant trends from 2017 to 2024. The overall ground subsidence increased from –3.74 mm in

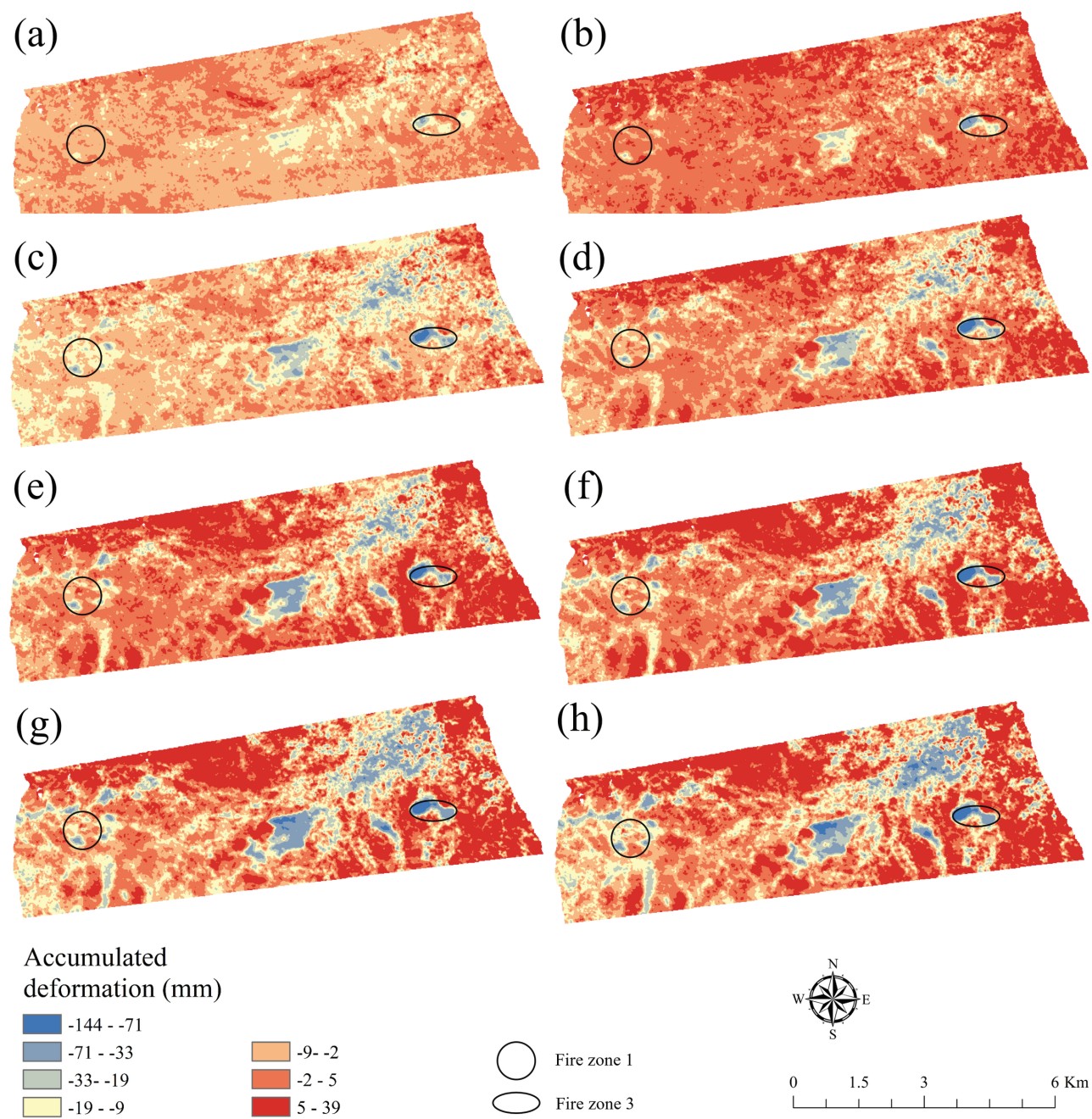

**Fig 6. The distribution map of surface subsidence.** (a), (b), (c), (d), (e), (f), (g), and (h) are 2017, 2018, 2019, 2020, 2021, 2022, 2023, and 2024, respectively (Republished from Desheng Xie under a CC BY license, with permission from Desheng Xie, original copyright 2024. The source data was from NASA Earth Observatory).

2017 to 1.70 mm in 2018, indicating a notable uplift, but then sharply declined to –7.32 mm in 2019, showing significant subsidence. It remained nearly stable, with a minimal change from 2023 to 2024. The ground subsidence in Fire Zone 1 also showed similar fluctuations. The ground subsidence in Fire Zone 3 was the most significant and continuous. It decreased from –28.01 mm in 2017 to –42.71 mm in 2018, showing notable subsidence. All surface subsidence from 2019 to 2024 exceeds –65 mm, indicating that the surface subsidence in Fire Zone 3 is more severe than that in the entire region.

## Discussion

### Analysis of fractional vegetation cover

Combined with the data provided, FVC fluctuated significantly from 2017 to 2024, especially in the low and lowest vegetation coverage. The decrease in the lowest vegetation cover area from 6.80 km$^2$ in 2017 to 6.10 km$^2$ in 2021 was attributed to successful policy interventions aimed at conservation and restoration, as well as containment of burning in coal fire zones, as evidenced by the reduction in active fire sites. However, the lowest vegetation coverage increased dramatically from 2021 onwards, reaching 10.27 km$^2$ in 2024. This was influenced by rising temperatures and lower humidity, which negatively affect-ed vegetation growth [31]. Medium vegetation cover remained relatively stable, reaching 7.32 km$^2$ in 2021 and declining slightly thereafter, indicating a more resilient vegetation base in these regions [32]. The higher vegetation cover areas decreased significantly from 2017 to 2022 and recovered to 3.43 km$^2$ in 2024. The highest vegetation coverage area had a small change range and maintained a stable change range, which indicated that the vegetation foundation was strong. Ecological restoration work was carried out in the early stage of the coalfield fire area, and various drought resistant and cold resistant grass species were planted, with good planting effects (Fig 7). Overall, vegetation covers experienced complex dynamic changes, which were influenced by a combination of environmental factors, policy measures, and coal fire zone management [33].

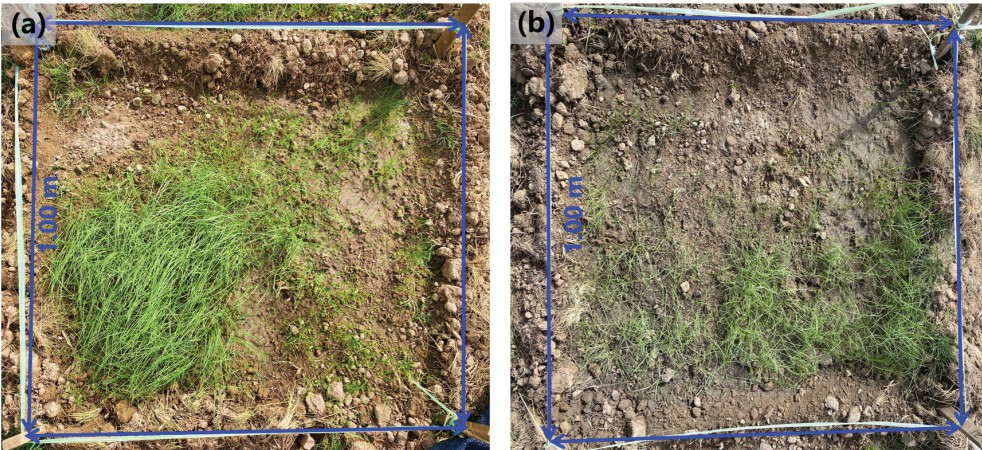

**Fig 7. The restoration effect carried out in the early stage of (a) Fire Zone 1 and (b) Fire Zone 3 (Republished from Desheng Xie under a CC BY license, with permission from Desheng Xie, original copyright 2024.** The source data was from NASA Earth Observatory).

In addition, FVC in Fire Zone 1 was significantly lower than overall FVC levels, highlighting the devastating impact of the fire on vegetation in the region and its slow recovery process. Although the FVC of Fire Zone 1 has been increasing year by year, its rate of in-crease has been slow and has not yet reached the overall average, indicating the need for a longer and more comprehensive recovery effort in the region. This lag may be because Fire Zone 1 experienced more severe fire damage and significant damage to the soil and vegetation base. In contrast, the FVC of Fire Zone 3 fluctuated during the observation period, but overall remained at a high level, and even exceeded the overall FVC in some years such as 2021. The rapid recovery of Fire Zone 3 may be due to its stronger natural resilience and less human disturbance [34]. In addition, the region may also benefit from more favorable environmental conditions, such as suitable soil types, adequate precipitation, and suitable temperatures, which are conducive to the growth and recovery of vegetation [35].

## Analysis of remote sensing ecological index

From the changes in RSEI data, the ecological environment quality fluctuated significantly between different years. This indicated changes in the ecological environment, influenced by various factors such as coal fire, climate change, and human intervention [36]. Fire Zone 3 maintained a high surface temperature, indicating the presence of a persistent heat source such as coal fire activity [37]. The overall wet remained relatively stable but showed a slight downward trend. It reached a lower point (0.445) in 2022, which could indicate worsening humidity conditions due to climate change, reduced water resources, or human activities [38]. Wet in Fire Zone 1 was relatively high between 2017 and 2019 but gradually decreased. This may be due to the evaporation of water caused by coal fire activity or a drop in the water table [39]. Affected by coal fire activity, the wetness of Fire Zone 3 was relatively low and fluctuated greatly during the whole observation period. Given the burning problem in coal Fire Zones, the government and relevant departments should take effective the coal fire monitoring and early warning system, implementing fire control projects, and carrying out ecological restoration [40]. With the implementation of the policy and the promotion of ecological restoration work, it is expected that the ecological environment quality will gradually improve. However, due to the complexity and uncertainty of coal-burning activities, continuous attention and monitoring of ecological environment changes in the region are necessary [41].

The poor-rated areas decreased from 8.471 $km^2$ in 2020 to 5.159 $km^2$ in 2021. It reflected the effectiveness of a series of ecological restoration measures during the period, and the improvement of climatic conditions (such as suitable temperatures and increased humidity) conducive to vegetation growth and ecological recovery [42]. However, the poor-rated areas had rebounded in subsequent years, especially in 2023. It revealed new pressures and challenges to the ecological situation. Poor ecological indices areas were more vulnerable to Fire Zone burning and extreme drought. On further observation, the medium-rated and good-rated areas remained relatively stable in most years. It might mean that despite the improvement of the overall ecological environment, and the protection of high-quality ecological areas still faces challenges [43]. Notably, there was a significant decrease in good-rated areas in 2023, which might be related to extreme climate events (such as heat and drought) or an increase in the intensity of human activities. The continuous high temperature and dry weather in a certain year would directly affect the growth of vegetation and soil moisture and harm RSEI. On the contrary, if the humidity increased, it was conducive to the growth and recovery of vegetation, which might raise the RSEI level [44,45].

## Analysis of flat spectral shape index

From the overall data of FSSI, although there were small fluctuations, showing the relative stability of coal reserves. However, several key years (such as 2017 and 2019) had seen FSSI decline, particularly to a low point of 0.619 in 2017, which might indicate an intensification of burning activity in the Fire Zone. Specifically, the FSSI of Fire Zone 1 was higher than that of Fire Zone 3 in most years, and the fluctuation was more significant. FSSI of Fire Zone 1 decreased significantly from 0.68 in 2020 to 0.65 in 2021, indicating that coal reserves in Fire Zone 1 decreased significantly during this period, most likely related to the intensification of burning activity [46–48]. In contrast, the change of FSSI in Fire Zone 3 was relatively stable, but it was still necessary to pay attention to its long-term trend [49,50].

At the same time, combined with the analysis of RSEI and FVC, it could be found that in the years when the FSSI decreased and the low-rated area increased, the decrease of RSEI indicated the increase of surface temperature in the Fire Zone, while the decrease of FVC indicated the deterioration of vegetation cover [51]. These composite indicators will collectively point to the intensification of combustion activities in coal mine Fire Zones and the reduction of coal mine reserves. To sum up, the continuous monitoring and comprehensive treatment of coal mine Fire Zones in this region were particularly important. It was suggested to strengthen the monitoring of Fire Zone 1 and Fire Zone 3, detect and deal with the burning activities in time, and take measures such as fire suppression and reclamation to restore the ecological environment and ensure the sustainable utilization of coal resources [52]. In addition, synergistic analysis with other environmental indicators needed to be strengthened to more fully assess the impact of coal mine Fire Zones and develop appropriate response measures [53].

## Analysis of land subsidence

The combustion of coal fires and the subsidence of the dump are the primary factors contributing to surface deformation. A detailed spatial analysis was performed to better understand the ground deformation caused by coal fire combustion and dump subsidence, as well as to analyze the spatial characteristics of deformation in Fire Zone 1 and 3 of the coalfields. Fig 8 depicts the surface deformation within the fire area of the coal-field. To present a clearer and more intuitive view of the spatial deformation patterns, Kriging interpolation was applied, and the interpolated results were converted into a three dimensional deformation map, as shown in Fig 8(c) and 8(e). To quantitatively analyze the spatial characteristics of surface deformation, L1 (Fire Zone 1) and L3 (Fire Zone 3), were drawn through both the coal fire area and the dump site. The surface deformation data along these sections for different years were extracted, and the results are shown in Fig 8(d) and 8(f).

The surface deformation in the coalfield fire area changes rapidly across space, with a complex overall deformation pattern. While the edge shapes are irregular, they follow certain discernible patterns. From Fig 8(d), the overall deformation pattern in Fire Zone 1 is complex, with multiple subsidence centers at varying degrees of development. A significant subsidence zone has formed, and by comparing it with high–resolution imagery of Fire Zone 1, a high level of alignment with the edge of the open–pit mine was observed. This alignment is likely due to surface deformation caused by the combustion of exposed coal seams at the edge of the mining area. The contour lines around these subsidence centers are relatively dense, with approximately equal spacing, indicating deeper depressions at the subsidence centers with steep surrounding gradients. The deformation direction of these subsidence centers varies with the spread of the coal fire and the intensity of combustion. The cumulative deformation

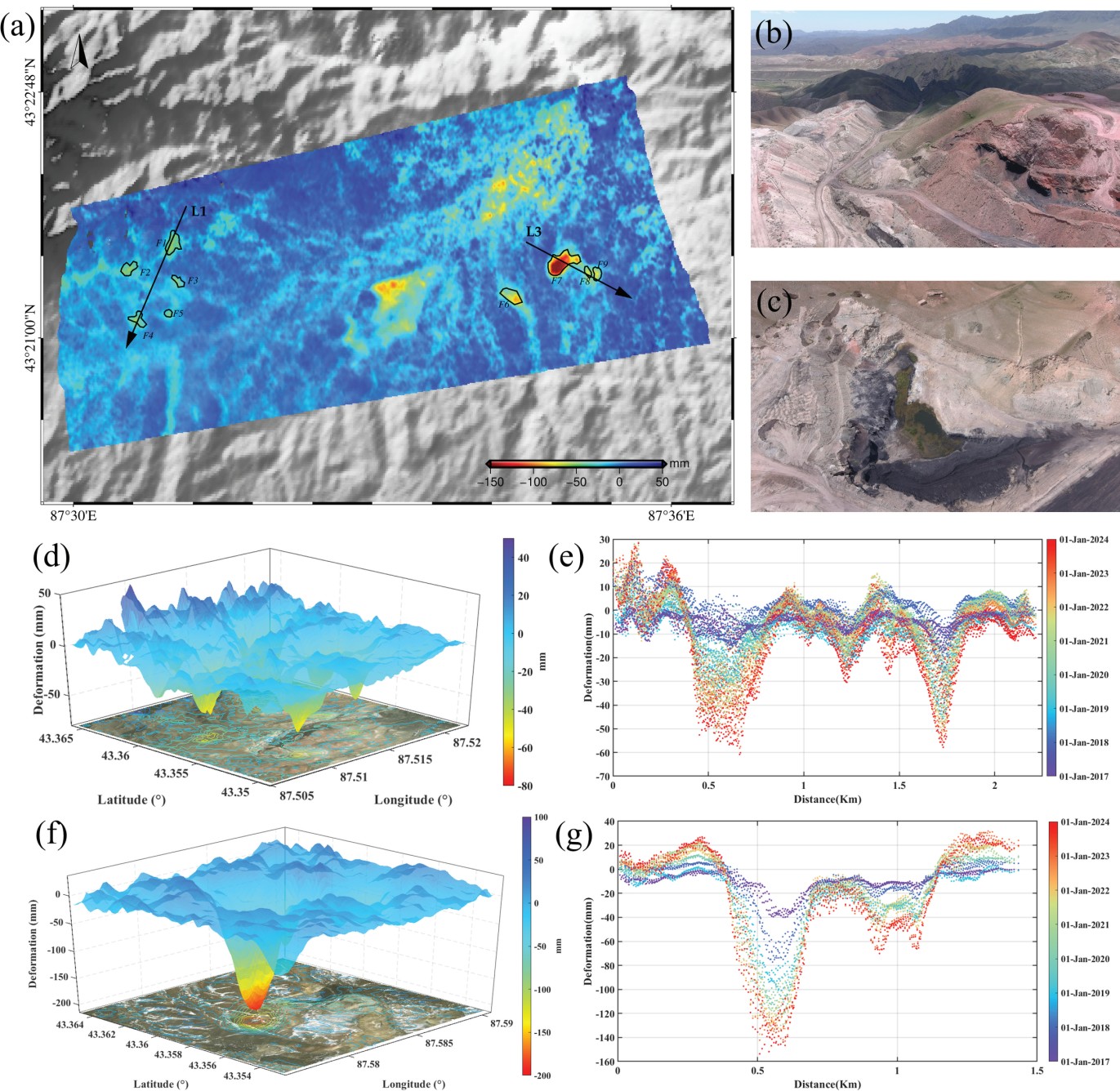

**Fig 8. Subsidence space (a), (b), (c), (d), (f), and curve (e), (g) in the mining area (Republished from Desheng Xie under a CC BY license, with permission from Desheng Xie, original copyright 2024.** The source data was from NASA Earth Observatory).

extracted along profile line L1 fluctuated significantly, showing multiple small subsidence centers, indicating extremely complex deformation within the subsidence zone.

There were obvious subsidence centers in Fire Zone 3, and the contours near these subsidence centers were dense and gradually thinning around them. The corresponding subsidence center depression was deep, gradually gentle around, and the edge was like the shape of

the dump. The main reason for this was that the burning in these areas was the floating coal after open–pit mining, rather than the coal seam at the edge of the open–pit mining area. L3 passes through the subsidence area caused by the subsidence of two dumps and the combustion subsidence of one fire area, where the accumulative shape variable of the most severe subsidence center reaches –157.23 mm. The annual cumulative shape variables of these settling regions are roughly equal. However, the cumulative subsidence from 2017 to 2019 was significant because the coal fire spread and burning intensity was higher during this period. This conclusion was also consistent with surface subsidence data.

## Analysis of surface temperature field

Based on the understanding and description of the above data, the deformation mechanisms in the coal Fire Zone were analyzed. The region primarily contained surface coal fires and coal pile fires, where the coal pile fires were less intense and caused minimal impact. Therefore, the focus was on studying the deformation forms and processes of the overlying rock layers and surface in the surface coal fire zones, providing reliable theoretical and technical support for preventive measures. Compared to Fire Zone 1, where the burning was less intense, Fire Zone 3 primarily burned outcropping coal seams. Thus, the study focused on the surface Fire Zone 3. Fig 9 shows the coal seam combustion and infrared thermography of Fire Zone 1 and

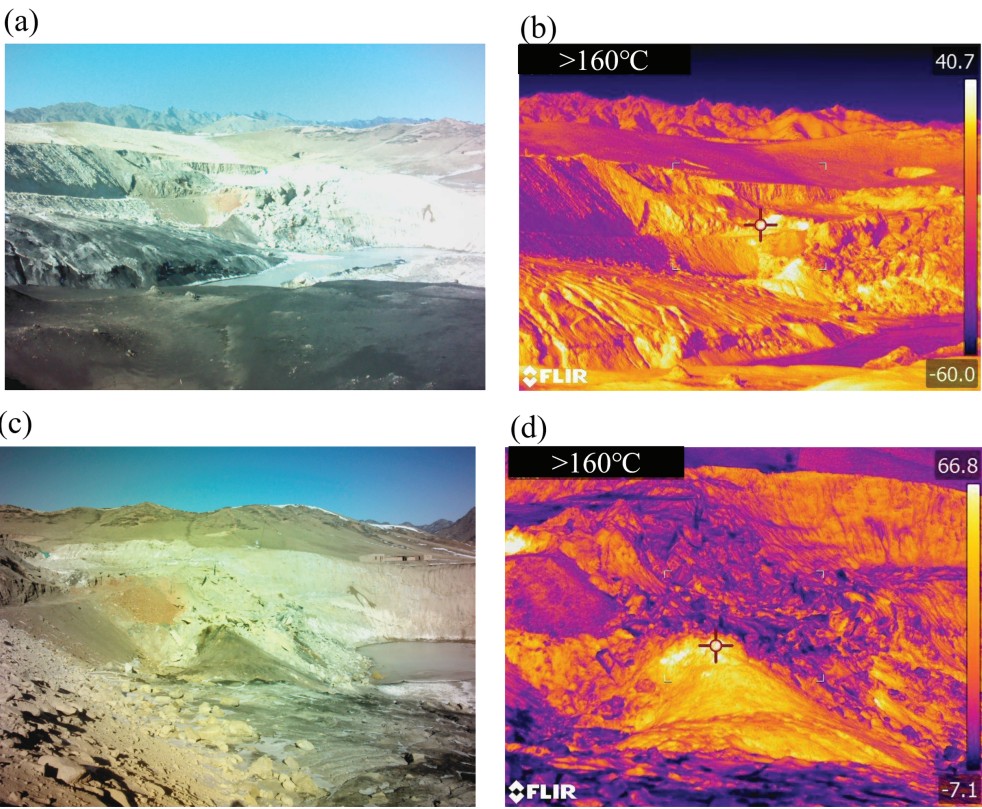

**Fig 9. Coal seam combustion (a), (c) and infrared thermometry map (b), (d) in Fire Zone 1 and 3 (Republished from Desheng Xie under a CC BY license, with permission from Desheng Xie, original copyright 2024. The source data was from NASA Earth Observatory).**

3. From the figure, it could be seen that during surface coal combustion, many cracks of varying degrees had formed, leading to surface subsidence and collapse. Additionally, the broken rocks formed by coal combustion covered the burning area, creating an insulated environment for heat retention (Fig 9c). The density difference between the air and smoke created fire wind pressure, driving the combustion, further intensifying the collapse of the overlying rock layers and surface.

Field investigations revealed strong northwesterly winds in the study area, which passed over the cracks where smoke escaped, enhancing fire wind pressure and intensify-ing the coal seam combustion in fire Zone 3. As the combustion continued, the damage to the overlying rock layers became increasingly complex and irregular, leading to continuous deformation, movement, slippage, and collapse over time and space, resulting in surface cracks and subsidence steps. When combustion reached a certain depth, the collapse of the overlying layers could not be immediately transmitted to the surface [54]. After combustion, an empty space formed below the overlying rock layers, resulting in three vertical zones: the collapse zone, fracture zone, and bending deformation zone. These zones were like the "three zones" in mining subsidence and were influenced by the depth of the burning coal seam, the area of the burning zone, and the mechanical properties of the overlying rock layers [55]. Along the direction of coal seam combustion, three horizontal zones formed: the burnout zone, combustion zone, and preheating zone. The burnout zone was composed of coal ash and collapsed rocks, which gradually compacted as the combustion center moved away [56]. In the combustion zone, coal fragments reacted with oxygen under high temperatures, but the viscous nature of the oxygen channels limited oxygen supply, causing the combustion to occur under oxygen-poor conditions. In the preheating zone, stress changes and heat released by combustion caused dehydration, drying, and coking, eventually leading to combustion, thus enabling the continuous burn-ing of the coal seam.

## Ecological impacts and recovery in coal fire zones

The dynamic changes of surface temperature, soil moisture, and nutrient content (TN, TP, and AP) in coal fires from 2017 to 2024 were analyzed to evaluate the impact of coal fires and control measures on the ecological environment (Fig 10). The mean surface temperature increased from 28.5 °C in 2017 to 34.2 °C in 2020 , and then gradually decreased to 21.3 °C in 2024 . The maximum decreased from 48.6 °C to 29.7 °C, indicating that the high temperature range of the fire area expanded in the early stage of treatment, which was subsequently reduced significantly as restoration efforts progressed. Soil moisture showed an initial increase in mean values from 32% in 2017 to 46% in 2020, reflecting the positive effects of irrigation. However, it declined sharply to 22% in 2024, with the maximum value dropping from 68% to 49%. This suggested that the persistent high temperatures and spread of fire zones caused substantial damage to soil structure despite early improvements.

TN content declined steadily from 52 mg/kg in 2017 to 11 mg/kg in 2024, indicating significant decomposition of soil organic matter due to high temperatures. Conversely, TP showed an increase in mean values from 195 mg/kg to 216 mg/kg by 2024, with the maximum value reaching 268 mg/kg. It indicated the effectiveness of localized fertilization measures. Available potassium (AP) decreased from 96 mg/kg in 2017 to 71 mg/kg in 2020 but rebounded to 118 mg/kg in 2024, with the maximum value significantly rising to 378 mg/kg, reflecting positive recovery effects in some areas.

Overall, coal fires have caused severe ecological damage, particularly to soil and vegetation. While control measures have achieved some improvements in specific indicators, sustained and long-term efforts remain essential for regional ecological restoration.

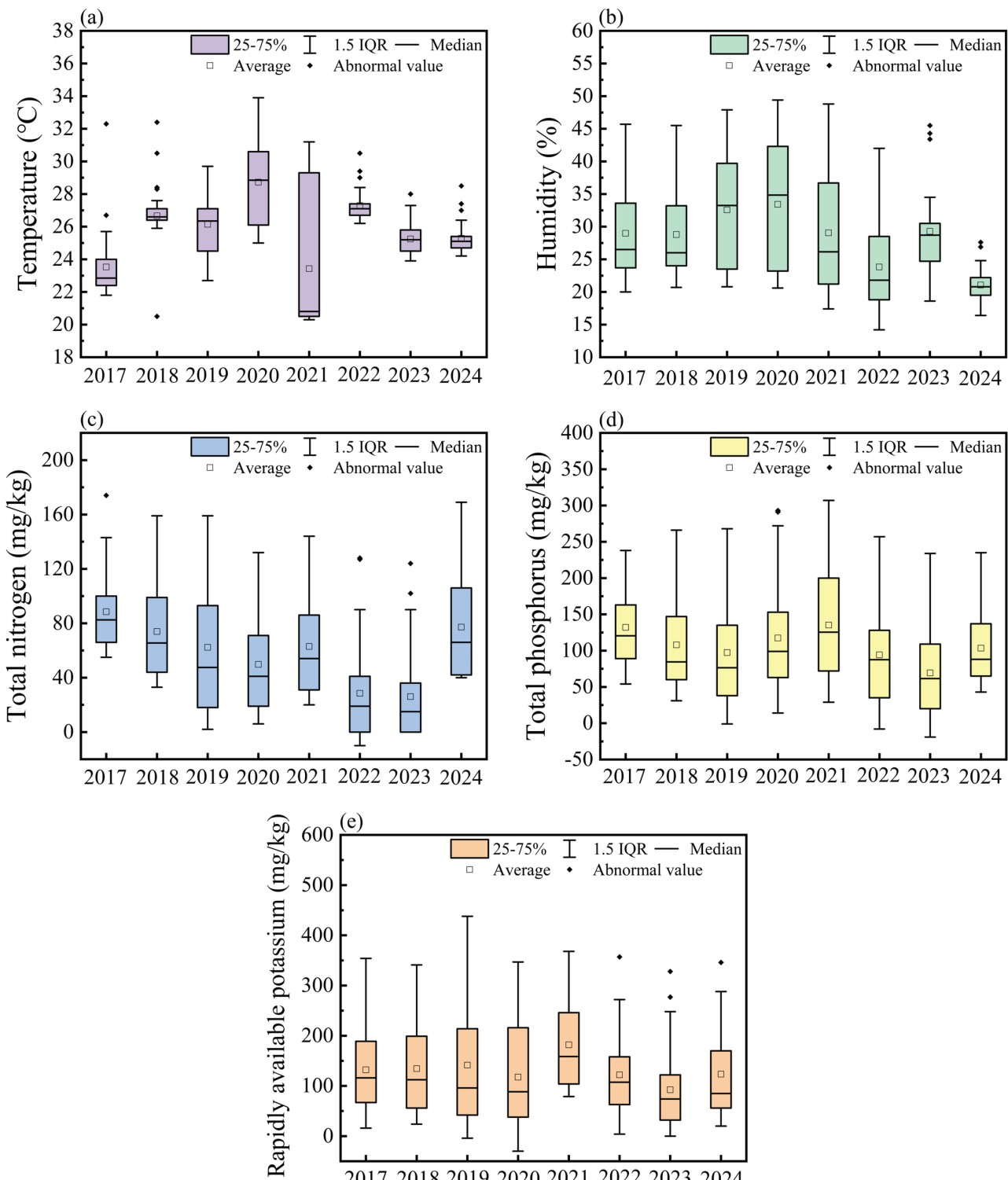

**Fig 10. Coal mine fire area sampling data. (a) temperature, (b) humidity, (c) total nitrogen, (d) total phosphorus, and (e) available potassium (Republished from Desheng Xie under a CC BY license, with permission from Desheng Xie, original copyright 2024.** The source data was from NASA Earth Observatory).

## Conclusions

This study integrates multi-source remote sensing data (FVC, RSEI, FSSI and land subsidence). The ecological environment changes and surface deformation of coal fire area in the southern mining area of Urumqi, Xinjiang during 2017-2024 were systematically evaluated. The main conclusions are as follows:

(1) The study found that temporary increases in FVC and RSEI from 2017 to 2021 were closely related to artificial interventions such as mine irrigation, soil improvement, and fire suppression works. However, the decline in the volatility of the ecological index after 2021 (FVC from 0.44 to 0.38) suggests that intermittent coal fires and an extremely dry climate offset some of the recovery effect. The results emphasize the importance of dynamic monitoring and adaptive management, and the need to adjust restoration strategies based on real-time remote sensing data.

(2) Although the initial ecological index of fire Zone 3 was high, its continuous high LST and annual land surface subsidence (<-65 mm) indicated that deep coal fire activity existed in the region, resulting in prolonged vegetation recovery cycle. In contrast, in fire Zone 1, the effectiveness of local fire suppression works in suppressing thermal anomalies is verified by surface covering and crack sealing measures. The research results provide a scientific basis for "zoning control" in coal fire area: thermal infrared remote sensing and InSAR technology should be prioritized for underground fire monitoring in fire area 3, while vegetation restoration and soil moisture can be emphasized in fire area 1.

(3) By FSSI and surface subsidence data, this study successfully identified the spatial coupling relationship between the spectral flat region (FSSI>0.7) and the subsidence center, and revealed the positive feedback mechanism of coal fire combustion, surface collapse and ecological degradation. This discovery can guide the design of "air-space-ground" integrated monitoring networks in mining areas, such as using FSSI to quickly locate exposed coal seams and verifying the location of fire points combined with UAV thermal imaging, thus improving the efficiency of fire fighting projects.

## Supporting information

**Source data**.
(ZIP)

## Author contributions

**Conceptualization:** Desheng Xie, Gensheng Li.

**Data curation:** Fantao Zeng.

**Funding acquisition:** Gensheng Li.

**Investigation:** Ke Wang.

**Methodology:** Desheng Xie.

**Software:** Baozhu Liu.

**Supervision:** Peng Liu.

**Validation:** Quan Fang.

**Visualization:** He Wu.

**Writing – original draft:** Yongwei Dong.

**Writing – review & editing:** Gensheng Li.

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
