## [Decision Letter · Decision Letter 0]

29 Jan 2025

PONE-D-24-58781Evaluating Land-Cover Change and Land Subsidence in Coal Fire Zones: Insights from Multi-Source MonitoringPLOS ONE

Dear Dr. Xie,

Thank you for submitting your manuscript to PLOS ONE. After careful consideration, we feel that it has merit but does not fully meet PLOS ONE’s publication criteria as it currently stands. Therefore, we invite you to submit a revised version of the manuscript that addresses the points raised during the review process.

We look forward to receiving your revised manuscript.

Kind regards,

Mikalai Filonchyk

Academic Editor

PLOS ONE

Journal Requirements:

This work was funded by the Basic Research Funds for Higher Education Institutions of Xinjiang Education Department (XJEDU2024J036), and Xinjiang Tianchi Talents (Young Doctor).

6. We note that your Data Availability Statement is currently as follows: All relevant data are within the manuscript and its Supporting Information files.

7. We note that Figures 1, 3, 4, 5, 6, and 8 in your submission contain [map/satellite] images which may be copyrighted. All PLOS content is published under the Creative Commons Attribution License (CC BY 4.0), which means that the manuscript, images, and Supporting Information files will be freely available online, and any third party is permitted to access, download, copy, distribute, and use these materials in any way, even commercially, with proper attribution. For these reasons, we cannot publish previously copyrighted maps or satellite images created using proprietary data, such as Google software (Google Maps, Street View, and Earth). For more information, see our copyright guidelines: http://journals.plos.org/plosone/s/licenses-and-copyright.

a. You may seek permission from the original copyright holder of Figures 1, 3, 4, 5, 6, and 8 to publish the content specifically under the CC BY 4.0 license.  

8. We note that Figures [2, 8, and 9] in your submission contain copyrighted images. All PLOS content is published under the Creative Commons Attribution License (CC BY 4.0), which means that the manuscript, images, and Supporting Information files will be freely available online, and any third party is permitted to access, download, copy, distribute, and use these materials in any way, even commercially, with proper attribution. For more information, see our copyright guidelines: http://journals.plos.org/plosone/s/licenses-and-copyright.

a. You may seek permission from the original copyright holder of Figures [2, 8, and 9]  to publish the content specifically under the CC BY 4.0 license. 

Reviewers' comments:

Reviewer's Responses to Questions

**Comments to the Author**

1. Is the manuscript technically sound, and do the data support the conclusions?

Reviewer #1: Yes

Reviewer #2: Yes

2. Has the statistical analysis been performed appropriately and rigorously? 

Reviewer #1: Yes

Reviewer #2: Yes

3. Have the authors made all data underlying the findings in their manuscript fully available?

Reviewer #1: Yes

Reviewer #2: Yes

4. Is the manuscript presented in an intelligible fashion and written in standard English?

Reviewer #1: Yes

Reviewer #2: Yes

5. Review Comments to the Author

Reviewer #1: The topic of this paper has important practical significance. It focuses on the surface coverage changes and land subsidence in coal fire areas, and reveals the ecological and geological effects of these areas through multi-source monitoring methods. The research method is innovative, which combines remote sensing, thermal infrared imaging, UAV survey and field investigation, and provides comprehensive data support for the monitoring of coal fire disaster. The topic selection has important practical significance, and the research method is scientific and reasonable, but there are still some improvements in some links, which can be accepted after modification.

1.It is suggested that the methods, parameters and basis of each link of data prepossessing should be described in detail to ensure the repeatability of data processing and the reliability of results, and to enhance the scientificity of research.

2.Although each monitoring index was analyzed separately, the correlation was not deep, and there was a complex feedback mechanism between vegetation cover change and soil nutrient, water and land subsidence. The paper did not fully explore these internal relations, and suggested that statistical analysis or model simulation methods could be used to study the interaction mechanism between vegetation, soil, topography and other factors.

3.The geological conditions, coal quality characteristics and climatic background of different coalfields are quite different, and the generalization of research conclusions is limited, and the basis for selecting Fire Zone 1 and Fire Zone 3 is not fully explained in the specific research background. It is suggested to add background introduction to explain the particularity and representativeness of the study area. If we can compare and analyze the data of other typical coal fire areas or refer to relevant research cases, and strengthen the discussion of regional universality, the application value of research results can be improved.

4.Figure 10 is located between figure 1 and figure 2, check whether the upload or sort error.

5.It is suggested that the conclusion part should be revised to clarify the practical application value and significance of the research.

Reviewer #2: The graphs and maps require quality improvements for better clarity. For instance, it is challenging to distinguish the gradations of indices (e.g., FVC and RSEI) in some figures.

Data processing methods, such as index normalization or interpolation, need more detailed descriptions. Additionally, it is unclear how the study zones (Fire Zone 1 and Fire Zone 3) were selected and why they were chosen for analysis.

Why not conduct a comparative analysis with other coal regions (e.g., India or the USA) to emphasize the uniqueness and scientific significance of the results? Adding an analysis of the influence of certain climatic and meteorological factors (e.g., changes in precipitation and temperature) on fire zones would improve the contextual understanding.

In the results section, the authors should clarify how changes in the RSEI and FVC indices are related to restoration measures. For example, how exactly did these measures impact vegetation recovery?

The authors mention the use of Landsat data but do not specify the resolution of the images or the frequency of their acquisition. This should be corrected.

Are potential remote sensing errors (e.g., atmospheric distortions or topographic effects) considered?

It is recommended to more clearly highlight the novelty of the study compared to previously published works, especially considering the use of multispectral data.

The text contains repetitions of key ideas, particularly in the results and discussion sections. For instance, it is repeatedly mentioned that coal fires lead to vegetation and soil degradation. Instead, the focus could be placed on the unique aspects of each study zone (Fire Zone 1 and Fire Zone 3), avoiding redundant repetition of already obvious conclusions.

Information about the methods used (e.g., the application of Landsat and UAVs) in the methodology section sometimes overlaps with the description of results. This creates an impression of duplication. It is recommended to keep the methodology in the appropriate section and the results in subsequent sections.

Some sections seem overly saturated with information. For instance, the discussion of FVC and RSEI contains numerous details that could be condensed into a table or presented graphically, allowing the text to focus solely on the interpretation of these data.

6. PLOS authors have the option to publish the peer review history of their article (what does this mean?). If published, this will include your full peer review and any attached files.

Reviewer #1: No

Reviewer #2: No

---

## [Author Response · Author response to Decision Letter 1]

11 Feb 2025

Responses to the editor's and the reviewers' comments

Reviewer #1:

Comment 1: It is suggested that the methods, parameters and basis of each link of data prepossessing should be described in detail to ensure the repeatability of data processing and the reliability of results, and to enhance the scientificity of research.

Response: Thank you for your comments. We have added some detailed parameters as requested. Additional relevant content has also been filled in on page 3.

“Landsat TM/ETM+/OLI/TIRS images from July 2013 to July 2024 were acquired from the United States Geological Survey (USGS) Earth Explorer platform. All images were selected with cloud cover ≤10% to minimize atmospheric interference. The FLAASH (Fast Line-of-sight Atmospheric Analysis of Spectral Hypercubes) module in ENVI was employed to correct for atmospheric scattering and absorption. Key parameters included: Mid-latitude winter atmospheric model (consistent with the study area’s semi-arid climate). Rural aerosol model with visibility set to 40 km. Water retrieval threshold was 1,130 nm. Images were co-registered to the WGS84/UTM Zone 45N coordinate system using ground control points (GCPs) from high-resolution Google Earth imagery (RMSE ≤ 0.5 pixels). Mosaics were generated using the "Seamless Mosaic" tool in ENVI, with histogram matching enabled to reduce inter-scene radiometric differences. Study area boundaries were clipped using a vector mask derived from the Xinjiang Geological Bureau’s regional maps.”

Comment 2: Although each monitoring index was analyzed separately, the correlation was not deep, and there was a complex feedback mechanism between vegetation cover change and soil nutrient, water and land subsidence. The paper did not fully explore these internal relations, and suggested that statistical analysis or model simulation methods could be used to study the interaction mechanism between vegetation, soil, topography and other factors.

Response: Thank you for your comments. There are indeed complex mechanisms between changes in vegetation cover and soil nutrients, moisture and land subsidence. However, due to the limitation of the length of the article, we cannot discuss these complex mechanisms. In this paper, the complex mechanism of surface subsidence, coal fire combustion and fire area restoration is discussed in depth. In the follow-up study, we will study the statistical model of these data based on your comments.

Comment 3: The geological conditions, coal quality characteristics and climatic background of different coalfields are quite different, and the generalization of research conclusions is limited, and the basis for selecting Fire Zone 1 and Fire Zone 3 is not fully explained in the specific research background. It is suggested to add background introduction to explain the particularity and representativeness of the study area. If we can compare and analyze the data of other typical coal fire areas or refer to relevant research cases, and strengthen the discussion of regional universality, the application value of research results can be improved.

Response: Thank you for your comments. Mining area restoration in other countries and related cases are of course the focus of our research and learning. However, due to the limitation of the length of the paper and the selection of suitable mining areas, we cannot add more detailed research and discussion in this paper to compare the differences between this research area and other mining areas. We will add the research of this content in the follow-up research. As for the influence of temperature and precipitation on the mining area, we have some discussion in the section of remote sensing ecological index. However, according to the LST and WET data, neither is a major factor. Therefore, we integrate the influence of temperature and precipitation on the mining area with other natural conditions, collectively referred to as the influence of natural conditions on the mining area.

As for the main limiting factors for the remediation of coal field fire areas, our judgment is caused by surface subsidence and the particularity of coal seams in this study area. The specific analysis paragraphs are as follows:

“Based on the understanding and description of the above data, the deformation mechanisms in the coal Fire Zone were analyzed. The region primarily contained surface coal fires and coal pile fires, where the coal pile fires were less intense and caused minimal impact. Therefore, the focus was on studying the deformation forms and processes of the overlying rock layers and surface in the surface coal fire zones, providing reliable theoretical and technical support for preventive measures. Compared to Fire Zone 1, where the burning was less intense, Fire Zone 3 primarily burned outcropping coal seams. Thus, the study focused on the surface Fire Zone 3. Figure 11 shows the coal seam combustion and infrared thermography of Fire Zone 1 and 3. From the figure, it could be seen that during surface coal combustion, many cracks of varying degrees had formed, leading to surface subsidence and collapse. Additionally, the broken rocks formed by coal combustion covered the burning area, creating an insulated environment for heat retention (Figure. 11c). The density difference between the air and smoke created fire wind pressure, driving the combustion, further intensifying the collapse of the overlying rock layers and surface.

Field investigations revealed strong northwesterly winds in the study area, which passed over the cracks where smoke escaped, enhancing fire wind pressure and intensifying the coal seam combustion in fire zone 3. As the combustion continued, the damage to the overlying rock layers became increasingly complex and irregular, leading to continuous deformation, movement, slippage, and collapse over time and space, resulting in surface cracks and subsidence steps. When combustion reached a certain depth, the collapse of the overlying layers could not be immediately transmitted to the surface. After combustion, an empty space formed below the overlying rock layers, resulting in three vertical zones: the collapse zone, fracture zone, and bending deformation zone. These zones were like the "three zones" in mining subsidence and were influenced by the depth of the burning coal seam, the area of the burning zone, and the mechanical properties of the overlying rock layers. Along the direction of coal seam combustion, three horizontal zones formed: the burnout zone, combustion zone, and preheating zone. The burnout zone was composed of coal ash and collapsed rocks, which gradually compacted as the combustion center moved away. In the combustion zone, coal fragments reacted with oxygen under high temperatures, but the viscous nature of the oxygen channels limited oxygen supply, causing the combustion to occur under oxygen-poor conditions. In the preheating zone, stress changes and heat released by combustion caused dehydration, drying, and coking, eventually leading to combustion, thus enabling the continuous burning of the coal seam.”

Comment 4: Figure 10 is located between figure 1 and figure 2, check whether the upload or sort error.

Response: Thank you for your comments. We fixed the error and resubmitted the high resolution paper map.

Comment 5: It is suggested that the conclusion part should be revised to clarify the practical application value and significance of the research.

Response: Thanks for your comments, we have revised the conclusion as requested. The revised conclusions are as follows: “This study integrates multi-source remote sensing data (FVC, RSEI, FSSI and land subsidence). The ecological environment changes and surface deformation of coal fire area in the southern mining area of Urumqi, Xinjiang during 2017-2024 were systematically evaluated. The main conclusions are as follows:

(1) The study found that temporary increases in FVC and RSEI from 2017 to 2021 were closely related to artificial interventions such as mine irrigation, soil improvement, and fire suppression works. However, the decline in the volatility of the ecological index after 2021 (FVC from 0.44 to 0.38) suggests that intermittent coal fires and an extremely dry climate offset some of the recovery effect. The results emphasize the importance of dynamic monitoring and adaptive management, and the need to adjust restoration strategies based on real-time remote sensing data.

(2) Although the initial ecological index of fire zone 3 was high, its continuous high LST and annual land surface subsidence (<-65 mm) indicated that deep coal fire activity existed in the region, resulting in prolonged vegetation recovery cycle. In contrast, in fire zone 1, the effectiveness of local fire suppression works in suppressing thermal anomalies is verified by surface covering and crack sealing measures. The research results provide a scientific basis for "zoning control" in coal fire area: thermal infrared remote sensing and InSAR technology should be prioritized for underground fire monitoring in fire area 3, while vegetation restoration and soil moisture can be emphasized in fire area 1.

(3) By FSSI and surface subsidence data, this study successfully identified the spatial coupling relationship between the spectral flat region (FSSI>0.7) and the subsidence center, and revealed the positive feedback mechanism of coal fire combustion, surface collapse and ecological degradation. This discovery can guide the design of "air-space-ground" integrated monitoring networks in mining areas, such as using FSSI to quickly locate exposed coal seams and verifying the location of fire points combined with UAV thermal imaging, thus improving the efficiency of fire fighting projects.”

Reviewer #2:

Comment 1: The graphs and maps require quality improvements for better clarity. For instance, it is challenging to distinguish the gradations of indices (e.g., FVC and RSEI) in some figures.

Response: Thanks for your comments, we have re-output the drawing form of the paper according to the requirements, and now the resolution of each paper drawing is guaranteed to be above 1000dpi.

Comment 2: Data processing methods, such as index normalization or interpolation, need more detailed descriptions. Additionally, it is unclear how the study zones (Fire Zone 1 and Fire Zone 3) were selected and why they were chosen for analysis.

Response: Thank you for your comments. We have added some detailed parameters as requested. Additional relevant content has also been filled in on page 3.

“Landsat TM/ETM+/OLI/TIRS images from July 2013 to July 2024 were acquired from the United States Geological Survey (USGS) Earth Explorer platform. All images were selected with cloud cover ≤10% to minimize atmospheric interference. The FLAASH (Fast Line-of-sight Atmospheric Analysis of Spectral Hypercubes) module in ENVI was employed to correct for atmospheric scattering and absorption. Key parameters included: Mid-latitude winter atmospheric model (consistent with the study area’s semi-arid climate). Rural aerosol model with visibility set to 40 km. Water retrieval threshold was 1,130 nm. Images were co-registered to the WGS84/UTM Zone 45N coordinate system using ground control points (GCPs) from high-resolution Google Earth imagery (RMSE ≤ 0.5 pixels). Mosaics were generated using the "Seamless Mosaic" tool in ENVI, with histogram matching enabled to reduce inter-scene radiometric differences. Study area boundaries were clipped using a vector mask derived from the Xinjiang Geological Bureau’s regional maps.”

The reasons for selecting fire areas are as follows: “There are three coal field fire areas in this study area, of which fire area 2 has been intensively treated and completely restored in 2015. However, due to the large area involved in fire zone 1 and fire zone 3 and the existence of open-pit coal mining work, restoration and mining have been parallel. In addition, the surface subsidence in these two areas has been relatively serious. Therefore, the selection of these two areas as key research areas for monitoring and restoration work is more in line with the working conditions of coal mines (parallel mining and restoration).”

Comment 3: Why not conduct a comparative analysis with other coal regions (e.g., India or the USA) to emphasize the uniqueness and scientific significance of the results? Adding an analysis of the influence of certain climatic and meteorological factors (e.g., changes in precipitation and temperature) on fire zones would improve the contextual understanding.

Response: Thank you for your comments. Mining area restoration in other countries and related cases are of course the focus of our research and learning. However, due to the limitation of the length of the paper and the selection of suitable mining areas, we cannot add more detailed research and discussion in this paper to compare the differences between this research area and other mining areas. We will add the research of this content in the follow-up research. As for the influence of temperature and precipitation on the mining area, we have some discussion in the section of remote sensing ecological index. However, according to the LST and WET data, neither is a major factor. Therefore, we integrate the influence of temperature and precipitation on the mining area with other natural conditions, collectively referred to as the influence of natural conditions on the mining area.

As for the main limiting factors for the remediation of coal field fire areas, our judgment is caused by surface subsidence and the particularity of coal seams in this study area. The specific analysis paragraphs are as follows:

“Based on the understanding and description of the above data, the deformation mechanisms in the coal Fire Zone were analyzed. The region primarily contained surface coal fires and coal pile fires, where the coal pile fires were less intense and caused minimal impact. Therefore, the focus was on studying the deformation forms and processes of the overlying rock layers and surface in the surface coal fire zones, providing reliable theoretical and technical support for preventive measures. Compared to Fire Zone 1, where the burning was less intense, Fire Zone 3 primarily burned outcropping coal seams. Thus, the study focused on the surface Fire Zone 3. Figure 11 shows the coal seam combustion and infrared thermography of Fire Zone 1 and 3. From the figure, it could be seen that during surface coal combustion, many cracks of varying degrees had formed, leading to surface subsidence and collapse. Additionally, the broken rocks formed by coal combustion covered the burning area, creating an insulated environment for heat retention (Figure. 11c). The density difference between the air and smoke created fire wind pressure, driving the combustion, further intensifying the collapse of the overlying rock layers and surface.

Field investigations revealed strong northwesterly winds in the study area, which passed over the cracks where smoke escaped, enhancing fire wind pressure and intensifying the coal seam combustion in fire zone 3. As the combustion continued, the damage to the overlying rock layers became increasingly complex and irregular, leading to continuous deformation, movement, slippage, and collapse over time and space, resulting in surface cracks and subsidence steps. When combustion reached a certain depth, the collapse of the overlying layers could not be immediately transmitted to the surface. After combustion, an empty space formed below the overlying rock layers, resulting in three vertical zones: the collapse zone, fracture zone, and bending deformation zone. These zones were like the "three zones" in mining subsidence and were influenced by the depth of the burning coal seam, the area of the burning zone, and the mechanical properties of the overlying rock layers. Along the direction of coal seam combustion, three horizontal zones formed: the burnout zone, combustion zone, and preheating zone. The burnout zone was composed of coal ash and collapsed rocks, which gradually compacted as the combustion center moved away. In the combustion zone, coal fragments reacted with oxygen under high temperatures, but the viscous nature of the oxygen channels limited oxygen supply, causing

---

## [Decision Letter · Decision Letter 1]

19 Mar 2025

Evaluating Land-Cover Change and Land Subsidence in Coal Fire Zones: Insights from Multi-Source Monitoring

PONE-D-24-58781R1

Dear Dr. Xie,

We’re pleased to inform you that your manuscript has been judged scientifically suitable for publication and will be formally accepted for publication once it meets all outstanding technical requirements.

Kind regards,

Mikalai Filonchyk

Academic Editor

PLOS ONE

Reviewers' comments:

Reviewer's Responses to Questions

**Comments to the Author**

1. If the authors have adequately addressed your comments raised in a previous round of review and you feel that this manuscript is now acceptable for publication, you may indicate that here to bypass the “Comments to the Author” section, enter your conflict of interest statement in the “Confidential to Editor” section, and submit your "Accept" recommendation.

Reviewer #1: All comments have been addressed

Reviewer #2: All comments have been addressed

2. Is the manuscript technically sound, and do the data support the conclusions?

Reviewer #1: Yes

Reviewer #2: Yes

3. Has the statistical analysis been performed appropriately and rigorously? 

Reviewer #1: Yes

Reviewer #2: Yes

4. Have the authors made all data underlying the findings in their manuscript fully available?

Reviewer #1: Yes

Reviewer #2: Yes

5. Is the manuscript presented in an intelligible fashion and written in standard English?

Reviewer #1: Yes

Reviewer #2: Yes

6. Review Comments to the Author

**Reviewer #1**: I am glad to receive this manuscript again. I noticed that the author has taken seriously and basically solved the problems raised in the last review process, and the quality of the manuscript has been greatly improved. The structure, logic and the presentation of experimental data have been significantly improved, and the combination of theoretical analysis and experimental results is closer. On the whole, the manuscript has basically met the requirements of the journal. Therefore, I recommend receiving the manuscript and look forward to its positive academic impact in related fields.

**Reviewer #2:** (No Response)

7. PLOS authors have the option to publish the peer review history of their article (what does this mean?). If published, this will include your full peer review and any attached files.

Reviewer #1: No

Reviewer #2: No

---

## [Editor Report · Acceptance letter]

PONE-D-24-58781R1

PLOS ONE

Dear Dr. Xie,

I'm pleased to inform you that your manuscript has been deemed suitable for publication in PLOS ONE. Congratulations! Your manuscript is now being handed over to our production team.

Kind regards,

on behalf of

Dr. Mikalai Filonchyk

Academic Editor

PLOS ONE